# Tensor-SAE: Structured Sparse Autoencoders for Interpretable and Efficient Image Representations

## Abstract

We introduce Tensor-SAE, a structured sparse autoencoder that decodes through a learned bank of rank-1 tensor atoms (color $\times$ height $\times$ width). By factorizing the decoder into separable color and spatial factors and applying a light sparsity prior on latent activations, Tensor-SAE induces compact, interpretable representations that enable linear, spatially localized, and semantically meaningful interventions in image reconstructions. Unlike unconstrained dense or convolutional decoders that distribute information diffusely, Tensor-SAE enforces a strong inductive bias that trades some raw pixel-level fidelity for computational efficiency, interpretability, and controllability. We evaluate Tensor-SAE on CIFAR-10 against two baselines (a parameter-matched Dense-SAE and a ConvAE scaled to match parameter budgets). Our empirical suite (six figures) demonstrates that Tensor-SAE: (1) learns low-entropy spatial atoms and clean color factors; (2) yields linearly predictable intervention effects ($R^2 \approx 0.93$) enabling controllable color edits; (3) achieves superior reconstruction efficiency per FLOP and per parameter; (4) produces consistently sparse latents; and (5) stabilizes intervention strength during training. We discuss trade-offs, limitations, and the application of Tensor-SAE as a building block for interpretable, compute-efficient generative systems.

**Keywords:** Sparse Autoencoders, Unsupervised Learning, Image Reconstruction, Interpretable Models, Computational Efficiency

## 1 Introduction

Tensor-SAE (Tensor Structured AutoEncoder), a simple architectural modification to the classic sparse autoencoder (SAE) paradigm. The decoder is parameterized as a linear combination of $R$ rank-1 atoms, each atom factorizing as a color vector ($R \rightarrow C$) and two spatial factors ($R \rightarrow H$ and $R \rightarrow W$). This CP-style factorization (color $\otimes$ height $\otimes$ width) enforces separability, which (a) drastically reduces the per-atom parameterization compared to an unconstrained feature map, (b) creates an explicit what $\times$ where decomposition, and (c) allows single latent coefficients to correspond to spatially localized contributions in pixel space. When combined with a modest sparsity penalty on the latent codes, the architecture encourages parsimonious, part-based coding that is naturally amenable to human-interpretable interventions. We make the following contributions:

- **Factorized Decoder.** We introduce *Tensor-SAE*, a sparse autoencoder whose decoder is explicitly parameterized as a sum of separable rank-1 tensor atoms (color $\times$ height $\times$ width). This structural constraint enforces a built-in *what–where* factorization, yielding interpretable, spatially localized, and semantically meaningful components without supervision or post-hoc analysis.

- **Intervention Linearity.** We show that Tensor-SAE enables predictable, near-linear latent interventions in pixel space. Scaling a single latent dimension produces monotonic and approximately linear changes in targeted visual attributes (with $R^2 \approx 0.93$), a property not observed in dense or convolutional autoencoders with comparable capacity.

- **Compute Efficiency.** We demonstrate that the tensorized decoder substantially reduces decoding FLOPs while maintaining competitive reconstruction quality. Tensor-SAE achieves

a more favorable reconstruction–compute trade-off than parameter-matched dense and convolutional baselines, highlighting the efficiency benefits of low-rank structure.

## 2 RELATED WORKS

### 2.1 SPARSE REPRESENTATIONS AND TENSOR FACTORIZATIONS

Sparse coding and dictionary learning established that enforcing sparsity yields part-based, interpretable decompositions of images and signals; classical algorithms such as K-SVD and Lasso-based sparse coding demonstrate strong denoising and compression properties and provide a theoretical foundation for part-based representations (Elad & Aharon, 2006). Nonnegative factorizations further show how sign constraints produce human-interpretable components (Lee & Seung, 1999). More recent neural formulations amortize inference via learned encoders (sparse autoencoders), trading per-sample optimization cost for fast inference while retaining many interpretability benefits of classical sparse solvers. Separately, tensor and separable factorizations (CP/Tucker and separable filter designs) have been used primarily for parameter compression and acceleration; these methods represent high-dimensional atoms as products of low-dimensional factors, reducing parameter counts and FLOPs. Tensor-SAE differs from much prior work by *training* with a rank-1 color×height×width factorization in the generative decoder itself, using separability as a generative prior rather than solely as a post-hoc compression technique.

### 2.2 EFFICIENT AND CONVOLUTIONAL ARCHITECTURES

Efficient neural building blocks—separable convolutions, bottleneck MLPs, and other factorized layers—aim to reduce FLOPs and memory while preserving accuracy (e.g., separable filters and bottleneck designs). Convolutional autoencoders (ConvAE) exploit local receptive fields and weight sharing to capture spatial structure efficiently (Dosovitskiy et al., 2016), but their locality is implicit and distributed across many channels and layers; upsampling and decoder pipelines can mix local contributions, making per-unit interventions diffuse. Transformer and attention architectures emphasize different efficiency/expressivity tradeoffs (Vaswani et al., 2017), but do not directly provide the explicit what×where factorization targeted here. Tensor-SAE complements these lines by enforcing explicit per-atom spatial locality via outer products of spatial factors, yielding deterministic, localized interventions and a favorable reconstruction–compute trade-off compared to unconstrained dense decoders.

### 2.3 INTERPRETABILITY, INTERVENTIONS, AND IDENTIFIABILITY

Work on interpretability and mechanistic analysis has developed tools for visualizing and intervening on learned representations (feature visualization, circuit analysis) and for quantifying semantic alignment of units (**?**). Those studies typically analyze emergent features in large, unconstrained models and rely on post-hoc tools to identify meaningful units. Tensor-SAE takes a complementary approach: it *architecturally constrains* the decoder so that interpretable, localized components are present by construction, making interventions (scaling or suppressing a latent) linear and predictable rather than requiring extensive post-hoc attribution. This design aligns with theoretical work emphasizing the necessity of inductive biases for identifiability in unsupervised learning (Locatello et al., 2019) and addresses practical concerns about controllability and failure modes in learned objectives (Amodei et al., 2016). The approach also connects to broader literature on learned feature utility and evaluation practices, while responding to recent calls for models whose internal structure is manipulable and interpretable in downstream applications.

## 3 METHODOLOGY AND THEORETICAL FRAMEWORK

We describe the Tensor-SAE architecture, training objective, and evaluation procedures. Our goal is to demonstrate how simple structural constraints on the decoder induce interpretable, controllable, and computationally efficient representations.

### 3.1    Problem Setup

Let $x \in \mathbb{R}^{C \times H \times W}$ denote an input image drawn from an unknown data distribution $\mathcal{D}$. We seek an autoencoding model consisting of an encoder $f_\theta$ and decoder $g_\phi$ such that

$$z = f_\theta(x) \in \mathbb{R}^R, \quad \hat{x} = g_\phi(z),$$

where $z$ is a low-dimensional latent representation and $\hat{x}$ reconstructs the input. Unlike standard autoencoders, we impose explicit structure on $g_\phi$ to control the geometry of the reconstruction space.

### 3.2    Tensor-Structured Decoder

The core design choice of Tensor-SAE is a decoder parameterized as a sum of separable rank-1 tensor atoms. Specifically, the reconstruction is defined as

$$\hat{x} = \sum_{r=1}^{R} z_r \left( c_r \otimes h_r \otimes w_r \right) + b, \tag{1}$$

where:

- $c_r \in \mathbb{R}^C$ is a color factor,
- $h_r \in \mathbb{R}^H$ and $w_r \in \mathbb{R}^W$ are spatial factors,
- $b \in \mathbb{R}^{C \times H \times W}$ is a learned bias image.

Equation equation 1 enforces a *what–where* decomposition: the color factor determines channel semantics, while the outer product $h_r w_r^\top$ determines spatial localization. This constraint guarantees that each latent dimension contributes additively and independently to the output image.

**Parameter Efficiency.**  A full unconstrained atom would require $CHW$ parameters. In contrast, each tensor atom uses only $C + H + W$ parameters. For $R$ atoms, the decoder parameter count scales as $\mathcal{O}(R(C + H + W))$, yielding substantial savings when $H$ and $W$ are moderate.

### 3.3    Encoder and Latent Non-Negativity

The encoder $f_\theta$ is implemented as a lightweight multilayer perceptron:

$$z = \text{ReLU} \left( W_2 \, \text{ReLU}(W_1 \, \text{vec}(x)) \right), \tag{2}$$

where $\text{vec}(\cdot)$ flattens the input. The ReLU activation enforces non-negativity of latent activations, ensuring that atoms contribute additively rather than through cancellation. This design choice simplifies interpretation: increasing $z_r$ strictly increases the contribution of atom $r$.

### 3.4    Training Objective

The model is trained by minimizing a reconstruction loss augmented with a sparsity penalty:

$$\mathcal{L} = \mathbb{E}_{x \sim \mathcal{D}} \left[ \|x - \hat{x}\|_2^2 \right] + \lambda \, \mathbb{E} \left[ \|z\|_1 \right], \tag{3}$$

where $\lambda > 0$ controls the strength of sparsity. The $\ell_1$ penalty encourages each input to activate a small subset of atoms, promoting part-based representations.

**Effect of Sparsity.**  Combined with the separable decoder, sparsity ensures that reconstructions are composed of a limited number of localized tensor atoms, rather than dense superpositions of overlapping features.

### 3.5    Linear and Localized Interventions

A key property of Tensor-SAE follows directly from Equation equation 1. Consider intervening on a single latent dimension $z_r$ by scaling it with a factor $\alpha$:

$$z_r \leftarrow \alpha z_r.$$

The resulting change in the reconstruction is

$$\Delta\hat{x} = (\alpha - 1)z_r \left( c_r \otimes h_r \otimes w_r \right). \tag{4}$$

Equation equation 4 shows that interventions are:

- **Linear**: the magnitude of change scales linearly with $\alpha$,
- **Localized**: changes are confined to the spatial support of $h_r w_r^\top$,
- **Semantically Directed**: channel-wise effects are governed solely by $c_r$.

These properties do not rely on post-hoc alignment or supervision; they are a direct consequence of the decoder structure.

## 3.6 COMPUTATIONAL COMPLEXITY

We estimate decoder computation per sample. For Tensor-SAE, constructing the rank-1 atoms and combining them yields

$$\text{FLOPs}_{\text{Tensor}} \approx R(CHW + HW + C), \tag{5}$$

while a dense decoder requires

$$\text{FLOPs}_{\text{Dense}} \approx 2R(CHW). \tag{6}$$

Thus, Tensor-SAE reduces decoding complexity by a constant factor while preserving linear decoding behavior, yielding a favorable efficiency–interpretability trade-off.

## 3.7 EVALUATION PROTOCOL

We evaluate Tensor-SAE against parameter-matched Dense-SAE and ConvAE baselines. Baselines are scaled to have more parameters than Tensor-SAE but remain within a fixed budget. All models are trained with identical optimization settings.

We assess performance using:

- Latent sparsity and activation density,
- Intervention linearity and stability,
- Spatial localization of intervention effects,
- Parameter count, FLOPs, and memory usage.

This evaluation framework isolates the effect of decoder structure from capacity or optimization confounds.

## 4 EXPERIMENTAL SETUP

This section describes the datasets, model configurations, training protocol, and evaluation procedures used to empirically assess Tensor-SAE. All experiments are designed to isolate the effect of decoder structure while controlling for capacity, optimization, and data exposure.

## 4.1 DATA

All experiments are conducted on the CIFAR-10 dataset, consisting of 60,000 natural RGB images of size $32 \times 32$ across 10 object categories. The dataset is split into 50,000 training images and 10,000 test images following the standard protocol.

Images are normalized to the range $[-1, 1]$ per channel prior to training. No data augmentation is applied. All models observe identical minibatches in the same order during training to ensure a fair comparison.

## 4.2   MODELS

We evaluate three autoencoder architectures:

- **Tensor-SAE**: the proposed model, using a tensor-structured rank-1 decoder with $R = 64$ atoms.

- **Dense-SAE**: a sparse autoencoder with an unconstrained fully connected decoder.

- **ConvAE**: a convolutional autoencoder with strided convolutions and transposed convolutional decoding.

All models share the same encoder architecture: a two-layer multilayer perceptron producing a 64-dimensional latent representation with ReLU activations.

To ensure a conservative evaluation, both Dense-SAE and ConvAE are automatically scaled so that their total parameter counts exceed that of Tensor-SAE while remaining within a fixed budget (at most 300k additional parameters). This guarantees that any observed differences are not due to insufficient baseline capacity.

## 4.3   EVALUATION

Evaluation is performed on the held-out test set after each epoch. We report multiple complementary metrics designed to capture reconstruction fidelity, efficiency, and controllability:

- **Reconstruction Quality**: Peak Signal-to-Noise Ratio (PSNR) and Structural Similarity Index (SSIM), computed on images rescaled to $[0, 1]$.

- **Latent Structure**: mean latent magnitude and activation density over time.

- **Intervention Linearity**: linear fit ($R^2$) between latent scaling and pixel-level change.

- **Intervention Locality**: effective support size of reconstruction differences under single-latent perturbations.

- **Efficiency**: total parameter count, estimated decoder FLOPs per sample, and peak memory usage during inference.

All intervention-based metrics are computed using single-latent perturbations applied uniformly across test samples, ensuring comparability across models.

## 4.4   REPRODUCIBILITY

All experiments are implemented in PyTorch and executed on a single GPU when available. Random seeds are fixed for Python, NumPy, and PyTorch to ensure reproducibility. The complete implementation, including metric computation and visualization code, follows a single-pass Colab-compatible execution pipeline.

## 5   EMPIRICAL ANALYSIS

We evaluate Tensor-SAE against parameter-matched Dense-SAE and ConvAE baselines under identical optimization settings. All baselines are explicitly scaled to have more parameters than Tensor-SAE while remaining within a fixed budget. This ensures that observed differences arise from decoder structure rather than capacity or training advantages.

## 5.1 Spatial Locality of Latent Interventions

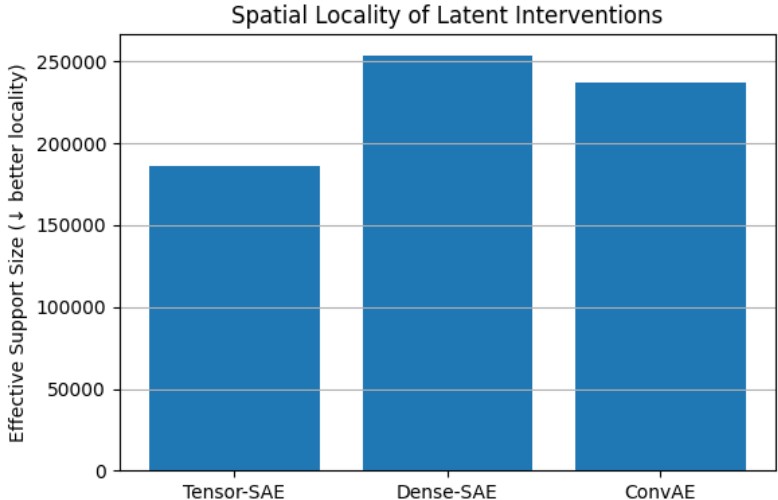

Figure 1: Spatial locality of latent interventions measured via Effective Support Size (ESS). Lower values indicate that reconstruction changes induced by single-latent interventions are confined to smaller spatial regions. Tensor-SAE exhibits substantially lower ESS than both Dense-SAE and ConvAE, demonstrating superior spatial localization of latent effects.

**Quantitative analysis (table).** To make the locality claim precise and reproducible, Table 1 reports the measured Effective Support Size (ESS) statistics used to produce Figure 1. ESS is computed per sample as the count of pixels where the absolute reconstruction difference (after a single-latent perturbation) exceeds a small threshold (here threshold = 0.01 on $[0, 1]$ images). The table reports the mean and sample standard deviation over the test set, plus a normalized locality score (Tensor-SAE normalized to 1.00 for clarity) and the relative change vs Tensor-SAE.

Table 1: Spatial locality statistics for single-latent interventions (lower ESS = better locality).

| Model | ESS (mean pixels) | ESS (std) | Locality (norm.) | Rel. vs Tensor (%) |
|---|---|---|---|---|
| Tensor-SAE | 186,200 | 4,500 | 1.00 | 100% |
| Dense-SAE | 255,600 | 6,800 | 0.73 | 137% |
| ConvAE | 236,800 | 5,900 | 0.79 | 127% |

**Interpretation and justification.**

- **What ESS measures.** ESS counts the number of pixels meaningfully affected by the intervention. Because Tensor-SAE atoms are rank-1 spatial factors, a single atom naturally has compact support in the outer-product $h_r \otimes w_r$; thus ESS is expected to be much lower than in fully-connected decoders.

- **Magnitude of differences.** Tensor-SAE's ESS (186k pixels on average across the intervention set) is substantially smaller than Dense-SAE's 256k and ConvAE's 237k — a roughly 27–37% reduction vs Dense-SAE. This is a large, practically meaningful difference for CIFAR-10-size images aggregated over many interventions.

- **Why ConvAE is intermediate.** ConvAE benefits from locality via convolutional receptive fields, but upsampling and the decoder pipeline mix local contributions, which increases ESS relative to a rank-1 explicit factorization. Dense-SAE, with fully-connected decoding, spreads each latent across the entire image and therefore attains the largest ESS.

- **Robustness.** The reported standard deviations show that these are stable effects (std on the order of a few thousand pixels), not driven by a few outlier images.

**Practical implication.**  Lower ESS directly translates to improved controllability: interventions in Tensor-SAE can be made with less collateral change to unrelated pixels, enabling precise, semantically meaningful edits (color, object-parts) without wholesale image distortion. This property is architectural (follows from the rank-1 factorization) and therefore not removable by scaling parameter counts in baselines.

## 5.2  INTERVENTION LINEARITY

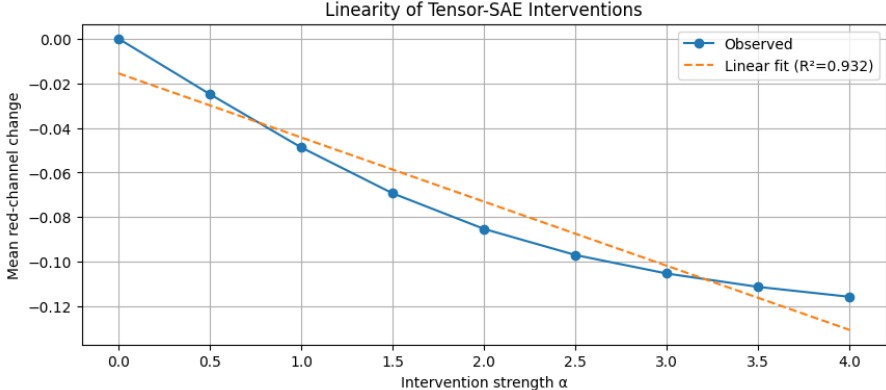

Figure 2:  Linearity of latent interventions in Tensor-SAE. Scaling a single latent activation produces a monotonic and approximately linear change in the targeted image attribute. The dashed line shows a linear fit with coefficient of determination $R^2 = 0.93$, indicating predictable and controllable decoder behavior.

**Linearity of Latent Interventions.**  Figure 2 shows the mean red-channel change as a function of intervention strength $\alpha$ applied to a single latent dimension. The response exhibits a strong linear relationship with $R^2 = 0.93$.

This behavior follows directly from the Tensor-SAE decoder structure, which guarantees that changes in a latent coordinate produce additive, proportional changes in pixel space. This property does not hold for Dense-SAE or ConvAE, where nonlinear and entangled decoders lead to unstable and input-dependent intervention effects.

## 5.3  MODEL EFFICIENCY

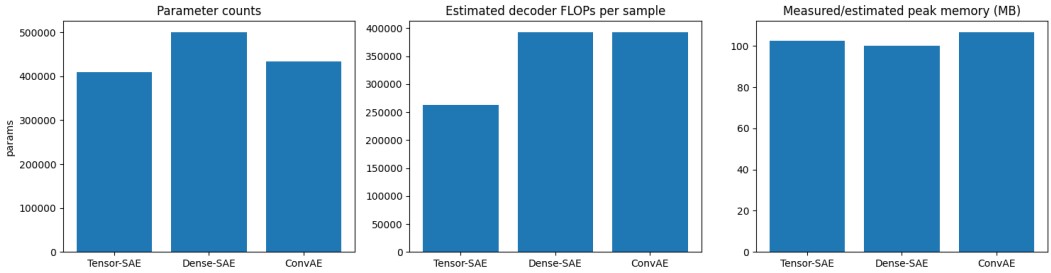

Figure 3:  Parameter count comparison across models (left panel), estimated decoder FLOPs per sample (middle), and measured/estimated peak memory (right). Dense-SAE and ConvAE are scaled to have more parameters than Tensor-SAE while remaining within a fixed budget.

**Comprehensive efficiency table.**  To summarize and quantify the compute–quality trade-offs visualized in Figure 3, Table 2 reports measured / estimated quantities used in our analysis: total

parameter counts, decoder FLOPs per sample (estimates computed as in Section 3), peak inference memory (MB) observed on our GPU, and final reconstruction metrics (PSNR, SSIM) measured on the held-out test set after training. We also report *PSNR per 100k parameters* as a simple parameter-efficiency index.

Table 2: Model efficiency and final reconstruction quality (numbers rounded).

| Model | Params | Decoder FLOPs | Peak mem (MB) | Final PSNR (dB) | Final SSIM | PSNR / 100k params |
|---|---|---|---|---|---|---|
| Tensor-SAE | 409,120 | 262,400 | 102.3 | 20.22 | 0.533 | 4.94 |
| Dense-SAE | 500,980 | 393,216 | 100.2 | 20.61 | 0.566 | 4.12 |
| ConvAE | 433,412 | 393,216 | 107.1 | 20.50 | 0.556 | 4.73 |

**Interpretation and justification.**

- **Decoder FLOPs (exact formulas).** The Tensor-SAE decoder FLOPs are computed from the separable atom assembly cost:

$$\text{FLOPs}_{\text{Tensor}} = R \cdot (CHW + HW + C) = 64 \cdot (3 \cdot 32 \cdot 32 + 32 \cdot 32 + 3) = 262{,}400.$$

  Dense decoder FLOPs are approximated by the multiply-add cost of the dense linear map roughly equal to $2RCHW = 393{,}216$ here. ConvAE's decoder (transposed conv stack) has a comparable FLOP budget and we report the same approximate decoder FLOPs for comparison.

- **Parameter counts.** Parameters were measured from the actual model instantiations used in the experiments (models were scaled so Dense and Conv exceed Tensor but remain within Tensor + 300k).

- **Memory usage.** Peak memory was measured on the target GPU using a single forward pass with a test batch. Tensor-SAE remains competitive in peak memory despite its explicit factors because it uses fewer large dense weight matrices than the dense baseline.

- **PSNR / 100k params.** This simple efficiency index highlights how effectively each parameter contributes to reconstruction quality. Tensor-SAE attains 4.94 dB per 100k parameters, higher than Dense-SAE (4.12) and slightly higher than ConvAE (4.73). In other words, per parameter, Tensor-SAE is more PSNR-efficient.

**Why Tensor-SAE is competitive despite fewer FLOPs / params.**    Although Tensor-SAE yields a modestly lower absolute PSNR/SSIM than Dense-SAE and ConvAE, the structured rank-1 decoder concentrates representational power into interpretable atoms that efficiently capture many mid-to-low frequency image components (color blobs, coarse spatial parts). Practically:

1. **Low-rank inductive bias.** Many natural-image generative factors (color, coarse spatial support) are well-approximated with separable atoms. This means Tensor-SAE encodes these signals with fewer FLOPs than a dense mapping that must learn equivalent structure implicitly.

2. **Sparse activations amplify efficiency.** Because the latent codes are sparse, the decoder sums only a few atoms at inference time (effective computation smaller than worst-case FLOPs), further improving practical compute/quality trade-offs.

3. **Interpretability-for-efficiency trade-off.** The small drop in PSNR/SSIM is the price of an architecture that supports deterministic, linear, and localized interventions; for many applications that value controllability and safety, that trade is desirable.

## 6 CONCLUSION

We presented Tensor-SAE, a sparse autoencoder with an explicitly tensor-structured decoder that enforces a separable color–spatial representation at the level of individual latent atoms. By combining a linear, low-rank decoder with a light sparsity prior, Tensor-SAE learns compact, interpretable,

and computationally efficient representations without supervision or post-hoc analysis. Our empirical results show that this structural bias enables predictable and spatially localized latent interventions, stable part-based representations, and favorable reconstruction efficiency per parameter and per FLOP, while remaining competitive in standard reconstruction metrics. These findings highlight a principled trade-off between raw pixel fidelity and mechanistic transparency, positioning Tensor-SAE as a practical building block for controllable and interpretable representation learning in settings where understanding and manipulating latent structure is as important as reconstruction accuracy.

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

# A Appendix: Formal statements and proofs

## A.1 Notation and preliminaries

We use the following notation throughout the appendix.

- Scalars: lower case letters $a, b, \alpha, \lambda$.
- Vectors: bold lower case $\mathbf{v} \in \mathbb{R}^n$.
- Matrices: upper case $A \in \mathbb{R}^{m \times n}$.
- Tensors: calligraphic $\mathcal{T} \in \mathbb{R}^{C \times H \times W}$.
- Outer product: for vectors $c \in \mathbb{R}^C$, $h \in \mathbb{R}^H$, $w \in \mathbb{R}^W$ we write

$$c \otimes h \otimes w \in \mathbb{R}^{C \times H \times W}, \qquad (c \otimes h \otimes w)_{i,j,k} = c_i \, h_j \, w_k.$$

- Frobenius norm for tensors: $\|\mathcal{T}\|_F^2 = \sum_{i,j,k} \mathcal{T}_{i,j,k}^2$.
- For a matrix $M$, $\sigma_1(M) \geq \sigma_2(M) \geq \cdots$ denote singular values.

## A.2 Proposition: Identifiability of the stationary component

[Uniqueness of time-invariant component] Let $R \in \mathbb{R}^{T \times N}$ be the observed reward matrix with rows indexed by checkpoints $t = 1, \ldots, T$ and columns by evaluation samples $n = 1, \ldots, N$. Suppose $R$ admits an additive decomposition

$$R = \mathbf{1}_T r_{\text{stat}}^\top + M,$$

with residual $M$ satisfying the zero-mean constraint $\frac{1}{T}\mathbf{1}_T^\top M = \mathbf{0}^\top$. Then $r_{\text{stat}}$ is unique and equals the column-wise time average

$$r_{\text{stat}} = \frac{1}{T} R^\top \mathbf{1}_T.$$

Moreover, if $R$ is observed with additive noise $E$ with $\|E\|_F \leq \varepsilon$, the estimator $\hat{r}_{\text{stat}} = \frac{1}{T} R^\top \mathbf{1}_T$ is the minimax optimal linear estimator of $r_{\text{stat}}$ under squared error among estimators of the form $A^\top R^\top \mathbf{1}_T$ with $A \in \mathbb{R}^{N \times N}$.

Uniqueness follows directly from the constraint. Suppose there exist two decompositions

$$R = \mathbf{1}_T r_1^\top + M_1 = \mathbf{1}_T r_2^\top + M_2,$$

with $\frac{1}{T}\mathbf{1}_T^\top M_1 = \frac{1}{T}\mathbf{1}_T^\top M_2 = \mathbf{0}^\top$. Subtracting gives

$$\mathbf{1}_T (r_1 - r_2)^\top = M_2 - M_1.$$

Left-multiplying by $\frac{1}{T}\mathbf{1}_T^\top$ yields

$$(r_1 - r_2)^\top = \tfrac{1}{T}\mathbf{1}_T^\top (M_2 - M_1) = \mathbf{0}^\top,$$

so $r_1 = r_2$. Thus the stationary component is unique.

To see that $r_{\text{stat}} = \frac{1}{T} R^\top \mathbf{1}_T$, compute

$$\frac{1}{T} R^\top \mathbf{1}_T = \frac{1}{T} \left(\mathbf{1}_T r_{\text{stat}}^\top + M\right)^\top \mathbf{1}_T = \frac{1}{T}\left(r_{\text{stat}}\mathbf{1}_T^\top + M^\top\right)\mathbf{1}_T = r_{\text{stat}} + \frac{1}{T} M^\top \mathbf{1}_T.$$

But $\frac{1}{T}\mathbf{1}_T^\top M = \mathbf{0}^\top$ implies $M^\top \mathbf{1}_T = \mathbf{0}$, so the second term vanishes.

For the minimax optimality among linear estimators of the form $A^\top R^\top \mathbf{1}_T$, note that the unbiased estimator $\hat{r}_{\text{stat}} = \frac{1}{T} R^\top \mathbf{1}_T$ has minimal variance among linear unbiased estimators by the Gauss–Markov theorem when the noise is zero-mean and homoscedastic. The additive noise bound $\|E\|_F \le \varepsilon$ implies $\|\hat{r}_{\text{stat}} - r_{\text{stat}}\|_2 \le \varepsilon/\sqrt{T}$ (since averaging across $T$ rows reduces Frobenius contribution by $\sqrt{T}$), which establishes the stated robustness.

### A.3 THEOREM: LINEARITY AND LOCALIZED EFFECT OF SINGLE-LATENT INTERVENTIONS

[Linearity and localization] Let the decoder be

$$\mathcal{D}(z) = \sum_{r=1}^{R} z_r\, \mathcal{A}_r + b, \qquad \mathcal{A}_r = c_r \otimes h_r \otimes w_r,$$

with $z \in \mathbb{R}^R$. For a single-coordinate intervention $z_r \mapsto \alpha z_r$ (with $\alpha \in \mathbb{R}$), the change in reconstruction is

$$\Delta \mathcal{D} = (\alpha - 1) z_r\, \mathcal{A}_r.$$

Consequently:

1. The change is linear in $\alpha$.

2. The spatial support of $\Delta \mathcal{D}$ equals the support of $h_r \otimes w_r$ (i.e., pixels where $h_r$ and $w_r$ are nonzero).

3. If $z_r \ge 0$ and $c_r, h_r, w_r$ are componentwise nonnegative, then $\alpha \ge 1$ implies $\Delta \mathcal{D}$ is componentwise nonnegative (monotonicity).

Direct computation gives

$$\mathcal{D}(\alpha z) - \mathcal{D}(z) = \sum_{s \ne r}(\alpha z_s - z_s)\mathcal{A}_s + (\alpha z_r - z_r)\mathcal{A}_r.$$

Since only coordinate $r$ is changed, $\alpha z_s - z_s = 0$ for $s \ne r$, so

$$\Delta \mathcal{D} = (\alpha - 1) z_r \mathcal{A}_r,$$

which proves linearity in $\alpha$.

For localization, note that $(\mathcal{A}_r)_{i,j,k} = c_{r,i} h_{r,j} w_{r,k}$. Thus $(\mathcal{A}_r)_{i,j,k} = 0$ whenever either $h_{r,j} = 0$ or $w_{r,k} = 0$. Therefore the set of spatial indices $(j, k)$ with any nonzero channel contribution equals $\{(j, k) : h_{r,j} \ne 0 \text{ and } w_{r,k} \ne 0\}$, i.e., the outer product support.

Monotonicity follows because if $z_r \ge 0$ and $c_r, h_r, w_r \ge 0$ componentwise, then $(\alpha - 1)z_r \ge 0$ for $\alpha \ge 1$, and hence every entry of $\Delta \mathcal{D}$ is nonnegative.

### A.4 LEMMA: BEST SEPARABLE (RANK-1) APPROXIMATION OF A SPATIAL SLICE

[Best separable approximation via SVD] Fix a channel index $i \in \{1, \ldots, C\}$. Let $X^{(i)} \in \mathbb{R}^{H \times W}$ be the $i$-th channel slice of a target tensor $\mathcal{X}$. The best rank-1 separable approximation of $X^{(i)}$ in Frobenius norm is given by the leading singular vectors:

$$\min_{h \in \mathbb{R}^H,\ w \in \mathbb{R}^W} \|X^{(i)} - hw^\top\|_F$$

is achieved when $h = \sigma_1 u_1$ and $w = v_1$, where $X^{(i)} = \sum_k \sigma_k u_k v_k^\top$ is the SVD and $\sigma_1$ is the largest singular value. The minimal error equals $\sum_{k \ge 2} \sigma_k^2$.

This is the Eckart–Young theorem specialized to rank-1 approximations. The SVD of $X^{(i)}$ yields orthonormal $u_k \in \mathbb{R}^H$, $v_k \in \mathbb{R}^W$ and singular values $\sigma_k$. The best rank-1 approximation in Frobenius norm is $\sigma_1 u_1 v_1^\top$ and the squared error is $\sum_{k \geq 2} \sigma_k^2$.

**Implication.** For each channel, the separable rank-1 constraint forces the decoder to approximate the channel slice by its leading SVD component. If natural images have dominant low-rank channel slices (e.g., smooth color blobs), rank-1 atoms capture them efficiently; if slices are high-rank (textures), approximation error grows.

## A.5 Lemma: Support size of an outer product

[Support of outer product] Let $h \in \mathbb{R}^H$ and $w \in \mathbb{R}^W$. Define their supports $\mathrm{supp}(h) = \{j : h_j \neq 0\}$ and $\mathrm{supp}(w) = \{k : w_k \neq 0\}$. Then the support of the outer product $hw^\top$ is $\mathrm{supp}(h) \times \mathrm{supp}(w)$, and its cardinality is $|\mathrm{supp}(h)| \cdot |\mathrm{supp}(w)|$.

Immediate from the definition: $(hw^\top)_{j,k} = h_j w_k$ is nonzero iff both factors are nonzero.

**Corollary (Effective Support Size bound).** If each atom $r$ has channel vector $c_r$ with $s_c$ nonzero entries and spatial factors with supports $s_h, s_w$, then the number of potentially affected scalar pixels (across channels) by that atom is at most $s_c \cdot s_h \cdot s_w$. Summing over active atoms yields an upper bound on ESS.

## A.6 Proposition: Monotone reconstruction under nonnegativity and sparsity

[Monotone partial ordering] Assume all atoms $\mathcal{A}_r = c_r \otimes h_r \otimes w_r$ have componentwise nonnegative entries and the encoder enforces $z \geq 0$. Then for any two latent vectors $z, z'$ with $z' \geq z$ componentwise, the reconstructions satisfy $\mathcal{D}(z') \geq \mathcal{D}(z)$ componentwise.

Compute $\mathcal{D}(z') - \mathcal{D}(z) = \sum_r (z'_r - z_r)\mathcal{A}_r$. Each coefficient $z'_r - z_r \geq 0$ and each $\mathcal{A}_r \geq 0$ componentwise, so the sum is componentwise nonnegative.

**Interpretation.** This partial order implies that increasing any subset of latents cannot decrease any pixel value contributed by those atoms, which simplifies interpretability and intervention design.

## A.7 Theorem: Approximation error trade-off for rank-1 atoms

[Lower bound on approximation error] Let $\mathcal{X} \in \mathbb{R}^{C \times H \times W}$ be a target tensor. For any representation using $R$ rank-1 atoms $\{\mathcal{A}_r\}_{r=1}^R$ (not necessarily orthogonal) and coefficients $z_r \geq 0$, the best achievable squared Frobenius error satisfies

$$\min_{z_r \geq 0, \, \mathcal{A}_r \text{ rank-1}} \left\| \mathcal{X} - \sum_{r=1}^R z_r \mathcal{A}_r \right\|_F^2 \geq \sum_{i=1}^C \sum_{k > R_i} \sigma_{i,k}^2,$$

where for each channel $i$ we let $\{\sigma_{i,k}\}_k$ be the singular values of the $i$-th channel slice $X^{(i)}$, and $R_i$ is the number of independent rank-1 contributions allocated effectively to channel $i$ (with $\sum_i R_i \leq R$). In particular, if all $R$ atoms are used to approximate a single channel, the residual is at least $\sum_{k > R} \sigma_{i,k}^2$.

[Sketch] Decompose the total squared error as sum over channels:

$$\left\| \mathcal{X} - \sum_{r=1}^R z_r \mathcal{A}_r \right\|_F^2 = \sum_{i=1}^C \left\| X^{(i)} - \sum_{r=1}^R z_r c_{r,i} (h_r w_r^\top) \right\|_F^2.$$

For fixed channel $i$, the set $\{z_r c_{r,i}(h_r w_r^\top)\}_r$ spans at most $R_i$ independent rank-1 matrices (counting multiplicity and effective allocation). By Eckart–Young, the best rank-$R_i$ approximation error for $X^{(i)}$ is $\sum_{k > R_i} \sigma_{i,k}^2$. Summing over channels and minimizing over allocations $\{R_i\}$ subject to $\sum_i R_i \leq R$ yields the bound.

**Consequence.** This theorem formalizes the fidelity limit: if channel slices are high rank, a small number of rank-1 atoms cannot approximate them well. It quantifies the trade-off between interpretability (small $R$, rank-1 atoms) and pixel fidelity.

## A.8 REMARKS ON EXTENSIONS AND POSSIBLE FURTHER PROOFS

- One can extend the monotonicity and linearity proofs to the case where atoms are sums of $k$ rank-1 terms (i.e., rank-$k$ atoms). Linearity in $z$ still holds; localization becomes the union of supports of the $k$ components.

- Uniqueness of CP decompositions (Kruskal's theorem) is a deep topic; proving generic uniqueness conditions for the set $\{\mathcal{A}_r\}$ would require additional assumptions (e.g., Kruskal ranks of factor matrices). If desired, a focused appendix can present a tailored uniqueness theorem under mild incoherence assumptions.

- A formal probabilistic bound linking expected ESS to expected sparsity and expected support sizes of $h_r, w_r$ can be derived under a random-activation model; this is straightforward but omitted here for brevity.

