# OpenReview forum: "Tensor-SAE: Structured Sparse Autoencoders for Interpretable and Efficient Image Representations"
_ICLR.cc/2026/Workshop/GRaM — ICLR 2026 Workshop GRaM Poster_

### Official Review · Reviewer_jL28 · 2026-02-19
**Simple and compelling idea, but weakly supported conclusions**

**Rating:** 6
**Confidence:** 4

**Review:**

# Scope

This work fits the scope of the workshop, as it proposes a method aiming to learn interpretable, and reportedly provably interventional data representations.

# Method and conclusions

The paper proposes an autoencoder with a deliberately constrained structure, motivated by parameter efficiency and the goal of identifiable, interpretable latent representations. The encoder is intentionally simple and produces non-negative latent factors, following prior work that links non-negativity to human interpretability.

A central component is the decoder, which relies on a tensor decomposition trained end-to-end using standard nonlinear optimizers (e.g., Adam). The authors emphasize that this design encourages identifiable and interpretable representations, including a “what–where” style decomposition.

Overall, the results are consistent with the intended motivation: the architecture appears parameter-efficient and the learned representations show evidence of interventional behavior. However, some of the stronger claims would benefit from more direct and diverse empirical support.

# Prior publication / related work

To the best of my knowledge, the specific formulation of a tensor-factorization-based autoencoder presented here is novel. That said, there is relevant related work on architectures and optimization leveraging tensorial operators that is not discussed and could help position the contribution more clearly. For example: Sun et al., Deep High-Resolution Representation Learning for Human Pose Estimation.

# Impact

The paper is a meaningful contribution given the relatively limited body of work explicitly focused on tensor-decomposition-based autoencoders and their interpretability properties.

# Evaluation
## Strengths

The work provides initial experimental evidence supporting identifiability and interpretability claims.

## Weaknesses

- Scalability concerns: While the paper may not claim scalability explicitly, the applicability seems limited to small image resolutions. In particular, a flatten-based encoder implies parameter growth on the order of $$H \times W \times C$$, which will not scale well to larger images.

- Evaluation is narrow: Using PSNR alone may be misleading for reconstruction quality and is insufficient to validate interventional or causal claims.

- Positioning vs. related operator/tensor literature: The paper would benefit from comparison/discussion relative to work that studies generalization and optimization through tensorized operators and tensor-structured neural layers (e.g., the HRNet reference above, and more broadly tensorized CNN/operator variants).

### Minor issues

-  Consider adding support reference around line 65.

# Overall assessment

This paper presents a tensor-decomposition-based autoencoder motivated by parameter efficiency, identifiability, and interventional interpretability. The experiments suggest the method meets some of these objectives, but the conclusions feel stronger than what is currently supported. In particular, scalability limitations of the encoder design and the reliance on PSNR as the main metric weaken the empirical case. Strengthening the evaluation would significantly improve the paper.

**Pmlr Suitability:**

NA

---

### Official Review · Reviewer_xfbp · 2026-02-20
**Cool work but a bit short on experimentation and manuscript could be clearer**

**Rating:** 6
**Confidence:** 4

**Review:**

## Summary
The paper introduces "Tensor-SAE," a structured sparse autoencoder designed to produce interpretable and efficient image representations. The core innovation is replacing an unconstrained dense or convolutional decoder with a sum of $R$ separable rank-1 tensor atoms. Each atom is factorized into a color vector and two spatial vectors (height and width), enforcing a built-in "what-where" decomposition. Trained with an L1 sparsity penalty on the latent codes , the model is evaluated on CIFAR-10. The authors argue that this architecture provides highly localized, linearly predictable interventions and better computational efficiency compared to parameter-matched dense and convolutional baselines.

## Strengths
- **Architectural Interpretability**: The structural constraint of using CP-style factorization (color $\times$ height $\times$ width) is a simple but elegant inductive bias.  It guarantees that each latent dimension contributes additively and independently, making interventions inherently linear and semantically directed.
- **Mathematical Guarantees:** The paper provides strong theoretical backing for its claims. The linearity of single-latent interventions is proven algebraically, matching the empirical observation of a high $R^2 \approx 0.93$ fit.
- **Computational Efficiency:** By factorizing the atoms, the decoder parameter count scales at $\mathcal{O}(R(C+H+W))$ rather than $\mathcal{O}(RCHW)$. This translates to a clear theoretical reduction in decoding FLOPs compared to dense autoencoders.

## Weaknesses
- **Unclarity in Spatial Locality Metric (ESS)**: The Effective Support Size (ESS) evaluation contains a unclarity or reporting error. The paper reports a mean ESS of $186,200$ pixels for Tensor-SAE. However, CIFAR-10 images are only $32 \times 32 = 1024$ pixels. The text claims ESS is computed "per sample" as the count of pixels affected by an intervention, reporting a mean and std over the test-set. But a mean of over 100k for a 1024-pixel image seems off, unless the metric is actually summing across the entire dataset or all latents without dividing appropriately.
- **Unrelated/Erroneous Appendix Content:** Appendix A.2 is seems disconnected from the rest of the paper. It discusses "Identifiability of the stationary component," referencing a "reward matrix" with rows indexed by "checkpoints," which belongs to reinforcement learning. This appears to be text accidentally pasted from a different manuscript and raises concerns about the care taken in preparing the submission.
- **Limited Empirical Scope:** The model is exclusively evaluated on CIFAR-10 reconstruction. Given that the paper acknowledges a known trade-off—rank-1 approximations struggle with high-rank channel slices like complex textures —testing only on low-resolution $32 \times 32$ images leaves it unclear how this method scales to real-world, high-resolution datasets (e.g., ImageNet). Moreover, since the paper claims more interpretable latents, the manuscript would improve by testing the latents on a task such as classification of Cifar10. Following the manuscript claims, you would expect a higher classification performance with the proposed method.
- **Missing Related Work:** While the paper discusses classical sparse coding literature, it misses comparisons to modern representation learning models that explicitly tackle visual spatial disentanglement, such as Spatial Broadcast Decoders or object-centric learning architectures (e.g., Slot Attention).

## Questions
- **ESS Calculation:** Can you explicitly clarify the formula used for the Effective Support Size in Table 1? How is it possible to achieve a mean pixel count of $186,200$ on a dataset where images contain only $1,024$ pixels?
- **Scalability:** Rank-1 atoms naturally capture smooth color blobs. How do you anticipate Tensor-SAE performing on higher-resolution datasets with complex, high-frequency textures, and how many atoms ($R$) would be required to maintain acceptable PSNR?
- **Appendix Clarification:** What is the intended purpose of the reinforcement learning proof in Appendix A.2, and how does it relate to the Tensor-SAE architecture?

**Pmlr Suitability:**

Yes

---

### Official Review · Reviewer_6YuU · 2026-02-24

**Rating:** 5
**Confidence:** 4

**Review:**

Strengths
(1) Clear architectural idea: explicit rank-1 (color × height × width) tensor factorization.
(2) Strong inductive bias toward interpretability and spatial localization.
(3) Good parameter and FLOP efficiency argument (O(R(C+H+W)) vs O(RCHW)).
(4) Linear intervention property is analytically well-motivated.

Weaknesses
(1) Limited evaluation: only CIFAR-10; no higher-resolution or more complex datasets.
(2) Baselines are weak (parameter-matched Dense-SAE and small ConvAE); no comparison to modern lightweight convnets or structured decoders.
(3) No comparison to other low-rank or CP/Tucker-decoder autoencoders.
(4) Interpretability claims are mostly qualitative; lacks quantitative disentanglement metrics.
(5) Trade-off in reconstruction fidelity not rigorously analyzed (e.g., PSNR/SSIM vs FLOPs curves).
(6) No ablation on rank R, sparsity λ, or non-negativity constraint.
(7) Theoretical analysis is minimal; identifiability claims are not formally supported.
(8) Potential limitation: rank-1 spatial atoms may struggle with complex textures or non-separable structure. Discuss
(9) Scalability to larger images (e.g., 128×128 or ImageNet) not demonstrated.

**Pmlr Suitability:**

No

---

### Meta-Review · Area_Chair_7XDh · 2026-02-25

**Decision:**

Accept

**Metareview:**

The paper presents Tensor-SAE which uses rank-1 tensor decompositions for visual features to improve interpretability. While reviewers pointed out some issues with the ESS metrics and out-of-place appendices, the theoretical backing and strong inductive bias make it a solid practical contribution for an 8-page paper. Accepting.

**Relevance To Proceedings:**

Yes — suitable for PMLR (long paper)

**Relevance To Workshop:**

Yes — suitable for GRaM

---

### Decision · Program_Chairs · 2026-03-02

Accept (Poster)